Association of red blood cell distribution width-platelet ratio with mortality after coronary artery bypass grafting

Zhang Bufan 1 2
Liu Yize 1
Zuo Jiyang 2 3
Song Tianxu 1
Wu Naishi wunaishi@tmu.edu.cn 1
1 Department of Cardiovascular Surgery, Tianjin Medical University General Hospital , Tianjin , China
2 Department of Cardiovascular Surgery, TEDA International Cardiovascular Hospital, Cardiovascular Clinical College of Tianjin Medical University , Tianjin , China
3 Department of Otorhinolaryngology Head and Neck Surgery, The Affiliated Lihuili Hospital of Ningbo University , Ningbo , Zhejiang , China
Leiherer Andreas
Electronic publication date: 2025 May 22
Publication date: 2025
Volume: 13
Electronic Location ID: e19472
Received 2024 Oct 14; Accepted 2025 Apr 24
Copyright: ©2025 Zhang et al.
Copyright year: 2025
Copyright holder: Zhang et al.
License: This is an open access article distributed under the terms of the Creative Commons Attribution License, which permits unrestricted use, distribution, reproduction and adaptation in any medium and for any purpose provided that it is properly attributed. For attribution, the original author(s), title, publication source (PeerJ) and either DOI or URL of the article must be cited.
License URL: https://creativecommons.org/licenses/by/4.0/

Keywords: Red blood cell distribution width-platelet ratio, In-hospital mortality, Coronary artery bypass grafting

Funding: Natural Science Foundation of Tianjin City 21JCYBJC00900 This work was funded by the Natural Science Foundation of Tianjin City (21JCYBJC00900). The funders had no role in study design, data collection and analysis, decision to publish, or preparation of the manuscript.

==============================
Background

This study aims to explore the association between red blood cell distribution width-platelet ratio (RPR) and mortality in patients after coronary artery bypass grafting (CABG).

Methods

Data on patients who underwent CABG from January 1, 2021, to July 31, 2022, were retrospectively collected. The locally weighted scatter plot smoothing (Lowess) method was utilized to display the crude association between RPR and in-hospital mortality. The areas under the receiver operating characteristic curves (AUC) were used to assess the discrimination. The cut-off value (0.107) of RPR was calculated using the Youden index method. The primary outcome was in-hospital mortality.

Results

In total, 1,258 patients were included. The Lowess curve showed an approximate positive linear relationship between RPR and in-hospital mortality. In the multivariable logistic regression model, RPR was an independent risk factor (OR 1.493, 95% CI [1.119–1.992] per standard deviation (SD) increase, p = 0.006) for in-hospital mortality after CABG. RPR (AUC 0.716, 95% CI [0.617–0.814]) demonstrated greater discrimination than RDW (AUC 0.578, 95% CI [0.477–0.680], p = 0.002). The cut-off value (0.107) of RPR was calculated for further analysis, and groups were further divided into the high RPR group (≥ 0.107) and the low RPR group (< 0.107). In the multivariable logistic regression model, high RPR (≥ 0.107) correlated with elevated risks of in-hospital mortality (OR 6.097, 95% CI [2.308–16.104], p < 0.001) and one-year mortality (OR 6.395, 95% CI [2.610–15.666], p < 0.001) after adjusting for all included covariates. Subgroup analyses revealed that high RPR consistently had increased risks of in-hospital mortality and one-year mortality. Besides, patients with low RPR show better one-year survival than those with high RPR.

Conclusion

Preoperative high RPR could serve as an independent risk predictor for in-hospital mortality and one-year mortality, which can be utilized to assess the prognosis of patients and further provide guidance for the treatment in patients following CABG.

Introduction

Coronary heart disease is common worldwide and the long-term therapeutic effect of coronary artery bypass grafting (CABG) has been confirmed. However, myocardial injury may occur in nearly a quarter of patients after CABG because of diverse mechanisms (Weidenmann et al., 2020). Postoperative adverse events following CABG still need concern and attention. Some research has reported the correlation between inflammation and immune-related indicators and adverse prognosis in cardiovascular diseases (Leiherer et al., 2016; Lin et al., 2024; Wasilewski et al., 2018). The red blood cell distribution width (RDW) has traditionally been utilized to diagnose anemia (Arkew et al., 2022). Previous studies have revealed that RDW has a strong association with inflammation, which is a vital contributor to coronary heart disease and can lead to plaque instability (Reed, Rossi & Cannon, 2017). The activation of the inflammatory response may affect the normal maturation of red blood cells by interfering with the erythrocyte membrane, leading to increased RDW (Lin et al., 2021; Peng, Sasmita & Luo, 2024). RDW has been identified as an independent prognostic indicator for individuals with myocardial infarction (MI) (Huang et al., 2021; Peng, Sasmita & Luo, 2024). Besides, platelets are involved in thrombosis and play a crucial role in the regulation of hemostasis and coagulation (Pezeshkpoor, Oldenburg & Pavlova, 2022). Abnormal platelet levels are associated with a higher occurrence of major adverse cardiovascular events in patients with acute coronary syndrome (Dahlen et al., 2021; Ito et al., 2018; Roh et al., 2020). Platelets are excessively activated and consumed under inflammatory conditions (Setarehaseman, Mohammadi & Maitta, 2025). Ischemia-reperfusion injury, swelling and hypoxia of microvascular endothelial cells and myocardial cells, microvascular spasm and embolism, the activation and release of inflammatory factors due to oxidative stress, and other factors are potential mechanisms for postoperative adverse events following CABG surgery (Chi et al., 2017; Tang et al., 2024). Elevated RDW and diminished platelet levels imply the concurrent presence of an inflammatory response and immune system dysfunction (Setarehaseman, Mohammadi & Maitta, 2025). Furthermore, recent research indicates that employing the ratio of RDW to platelet as a prognostic indicator can markedly enhance prognostic value when compared to utilizing each indicator in isolation (Li et al., 2019). The red blood cell distribution width-platelets ratio (RPR), as a combination parameter, has been reported as a new indicator to reflect the severity of inflammation (Tang et al., 2024). It has been shown to have a strong association with the prognosis of cancer, ischemic stroke, heart failure, and hepatic diseases (Dallio et al., 2024; Lin et al., 2024; Takeuchi et al., 2017; Xu & Peng, 2023). The role of RPR on the prognosis of patients after CABG still remains unclear gaps to elucidate. Thus, this study aims to explore the association between RPR with mortality in patients after CABG and provide a theoretical basis for predicting clinical outcomes after CABG.

Materials & Methods

Study population

Data on patients with CABG were retrospectively collected from January 1, 2021, to July 31, 2022, at TEDA International Cardiovascular Hospital. In this study, patients were selected in terms of the following criteria: (1) no history of cardiac surgery or sternotomy; (2) receiving CABG solely and admission for the first time; (3) grafts were the internal mammary artery and(or) great saphenous vein; (4) no infection or cancer comorbidities. Exclusion criteria were as follows: (1) received valve surgery or great vessel surgery; (2) receiving radiofrequency ablation; (3) use of implantable cardioverter-defibrillator; (4) undergoing emergent CABG due to cardiogenic shock; (5) incomplete data of RDW or platelets; (6) loss of follow-up within one year. This study was approved by the Institutional Review Board of TEDA International Cardiovascular Hospital (No. 2022-0802-1). All the information was anonymized to protect patients’ privacy so informed consent was waived by the Institutional Review Board of TEDA International Cardiovascular Hospital. Besides, the study was conducted in accordance with the ethical guidelines of the Helsinki Declaration.

Data selection and outcome definition

The following data were collected as baseline characteristics: age, gender, body mass index (BMI), mean arterial pressure (MAP), New York Heart Association (NYHA) classification, history of previous MI, and percutaneous coronary intervention (PCI), and stroke, number of stenosed arteries, number of grafts, EuroSCORE, and use of cardiopulmonary bypass. The evaluation of the NYHA classification for each patient was routinely assessed on admission according to European Society of Cardiology Guidelines for the diagnosis and treatment of acute and chronic heart failure (McMurray et al., 2012). Laboratory parameters included white blood cell (WBC), platelet, RDW, hemoglobin, RPR, serum creatinine, blood urea nitrogen (BUN), serum albumin, creatine kinase-myocardial band (CK-MB), Troponin I, and B-type natriuretic peptide (BNP). All the laboratory parameters were collected when the patients were admitted to the hospital. Platelet, WBC, RDW, and hemoglobin were detected by the fully automatic blood cell analyzer (manufacturer: Sysmex). Serum creatinine, BUN, and serum albumin were detected by the fully automatic biochemical analyzer (manufacturer: Sysmex). Troponin I, CK-MB, and BNP were detected by the fully automatic chemiluminescence immunoassay analyzer (manufacturer: Sysmex). Echocardiography was used to detect cardiac function, including left ventricular ejection fraction (LVEF), mild to moderate mitral regurgitation (MR), tricuspid regurgitation (TR), aortic regurgitation (AR), and pulmonary arterial hypertension (PAH). SYNTAX scores were obtained from the coronary angiogram. Comorbidities included peripheral artery disease, hyperlipemia, chronic obstructive pulmonary disease (COPD), and diabetes. The primary outcome was defined as in-hospital mortality.

Statistical analysis

Categorical data were shown as numbers and percentages, and compared using Pearson’s chi-squared test or Fisher’s exact test. Continuous data were demonstrated as median with interquartile range (IQR), and compared using Student’s t-test or Wilcoxon rank-sum test as appropriate. The locally weighted scatter plot smoothing (Lowess) method was utilized to display the crude association between RPR and in-hospital mortality. We completed the receiver operating characteristic (ROC) curves and calculated the area under the receiver operating characteristic curves (AUC) to assess the discrimination. The Youden index method was used to calculate the cut-off value of RPR. For a better explanation, groups were divided according to the cut-off value. The Kaplan–Meier method with log-rank test was performed to compare the one-year survival rates between high and low RPR. The day for each patient undergoing CABG has been regarded as the start of calculating the one-year follow-up. Subgroup analyses were performed to detect the relationship between RPR and in-hospital and one-year mortality among various groups according to median age, gender, median SYNTAX score, median MAP, median RDW, median BNP, and median LVEF. The statistical analyses and data visualizations were performed using SPSS software version 26.0 (IBM Corp., Armonk, NY, USA), Stata version 16.0, and R software version 4.0.2. P value < 0.05 with a two-sided test was identified as statistical significance.

Results

Baseline characteristics

A total of 1,258 patients were included in our study, and the incidence of in-hospital mortality was 3.58%, which has been detailed in Table 1. In the whole cohort, the median age (IQR) of included patients was 62 years (57, 66) and 56.92% were male. A total of 52.46% of patients have a history of MI, and 26.07% of patients underwent PCI previously. In this study, 536 (42.61%) patients underwent on-pump CABG, and 722 (57.39%) patients underwent off-pump CABG.

Table 1 Baseline characteristics in the whole cohort.

Variables	Whole Cohort (n = 1258)	
Age (years)	62 (57, 66)	
Gender, male, n (%)	716 (56.92)	
BMI (kg/m2)	25.85 (21.90, 29.68)	
MAP (mm Hg)	97.33 (86.67, 108)	
NYHA, n (%)		
I	13 (1.03)	
II	905 (71.94)	
III	303 (24.09)	
IV	37 (2.94)	
Previous MI, n (%)	660 (52.46)	
Previous PCI, n (%)	328 (26.07)	
Previous stroke, n (%)	149 (11.84)	
Number of stenosed arteries	3 (3, 3)	
Number of grafts	4 (3, 4)	
SYNTAX score	31 (25, 35)	
EuroSCORE	4 (3, 5)	
Cardiopulmonary bypass, n (%)	536 (42.61)	
WBC (K/μl)	7 (5.8, 8.1)	
Platelet (K/μl)	198 (155, 254.25)	
RDW (%)	14.5 (13.6, 15.9)	
Hemoglobin (g/L)	138 (134, 142)	
RPR	0.08 (0.06, 0.10)	
Creatinine (umol/L)	64 (58, 70)	
BUN (mmol/L)	6.7 (5.9, 7.6)	
Albumin (g/L)	43 (38, 47)	
CK-MB (U/L)	12.5 (10, 15)	
Troponin I (ng/ml)	0.28 (0.06, 1.07)	
BNP (pg/ml)	267.5 (98, 713.75)	
LVEF (%)	46 (40, 51)	
Mild to moderate MR, n (%)	196 (15.58)	
Mild to moderate AR, n (%)	97 (7.71)	
Mild to moderate TR, n (%)	49 (3.90)	
Mild to moderate PAH, n (%)	132 (10.49)	
Peripheral artery disease, n (%)	235 (18.68)	
Hyperlipemia, n (%)	128 (10.17)	
COPD, n (%)	171 (13.59)	
Diabetes, n (%)	598 (47.54)	
In-hospital mortality, n (%)	45 (3.58)	
One-year mortality, n (%)	59 (4.69)	
Notes.

Abbreviations AR aortic regurgitation

BMI body mass index

BNP B-type natriuretic peptide

BUN blood urea nitrogen

CK-MB creatine kinase-myocardial band

COPD chronic obstructive pulmonary disease

LVEF left ventricular ejection fraction

MAP mean arterial pressure

MI myocardial infarction

MR mitral regurgitation

NYHA New York Heart Association

PAH pulmonary arterial hypertension

PCI percutaneous coronary intervention

RDW red blood cell distribution width

RPR red blood cell distribution width-platelet ratio

TR tricuspid regurgitation

WBC white blood cell

Association between RPR and in-hospital mortality

Univariate logistic regression analyses showed that platelet (odds ratio [OR] 0.361, 95% CI [0.236–0.553] per standard deviation [SD] increase, p < 0.001), RDW (OR 1.539, 95% CI [1.217–1.945] per SD increase, p < 0.001), RPR (OR 1.741, 95% CI [1.485–2.041] per SD increase, p < 0.001), BNP (OR 1.290, 95% CI [1.006–1.655] per SD increase, p = 0.045), LVEF (OR 0.708, 95% CI [0.519–0.966] per SD increase, p = 0.029), previous stroke (OR 2.520, 95% CI [1.248–5.088], p = 0.010), and EuroSCORE (OR 1.360, 95% CI [1.019–1.815] per SD increase, p = 0.037) were accountable for in-hospital mortality. The details of factors selected by univariable logistic regression analyses were presented in the Supplemental Information 1. Moreover, RDW (OR = 1.354, 95% CI [1.010–1.815] per SD increase, p = 0.042) and RPR (OR 1.493, 95% CI [1.119–1.992] per SD increase, p = 0.006) were independent risk predictors by multivariate logistic regression analysis (Table 2). As shown in Fig. 1, the Lowess curve showed an approximate positive linear relationship between RPR and in-hospital mortality after CABG.

Table 2 Logistic regression assessing risk factors for in-hospital mortality.

Variables	Univariate analysis	Multivariate analysis	
	OR (95% CI)	p value	OR (95% CI)	p value	
Platelet	0.361 (0.236, 0.553)	<0.001	0.724 (0.409, 1.281)	0.267	
RDW	1.539 (1.217, 1.945)	<0.001	1.354 (1.010, 1.815)	0.042	
RPR	1.741 (1.485, 2.041)	<0.001	1.493 (1.119, 1.992)	0.006	
BNP	1.290 (1.006, 1.655)	0.045	1.228 (0.939, 1.607)	0.134	
LVEF	0.708 (0.519, 0.966)	0.029	0.724 (0.520, 1.009)	0.057	
EuroSCORE	1.360 (1.019, 1.815)	0.037	1.280 (0.937, 1.748)	0.121	
Previous stroke	2.520 (1.248, 5.088)	0.010	2.141 (0.987, 4.643)	0.054	
Notes.

Abbreviations BNP B-type natriuretic peptide

CI confidence interval

LVEF left ventricular ejection fraction

OR odds ratio

RDW red blood cell distribution width

RPR red blood cell distribution width-platelet ratio

WBC white blood cell

Figure 1 Association of red blood cell distribution width-platelet ratio and in-hospital mortality.

The crude association was displayed using the locally weighted scatter plot smoothing (Lowess) method. The shaded area represents the 95% confidence interval. Data were collected from individual human samples.

ROC analysis

ROC analyses were performed according to the selected predictors by multivariable logistic regression analysis. RPR (AUC 0.716, 95% CI [0.617–0.814]) showed better discrimination than RDW (AUC 0.578, 95% CI [0.477–0.680], p = 0.002) for the in-hospital mortality among patients after CABG (Fig. 2). The cut-off value (0.107) of RPR was calculated for further analysis, and groups were further divided into the high RPR group (≥ 0.107) and the low RPR group (< 0.107). The details were depicted in Table 3. The incidence of in-hospital and one-year mortality were both significantly higher in the high RPR group (13.02%, p < 0.001; 18.60%, p < 0.001), as compared to the low RPR group (1.63%; 1.82%).

Figure 2 ROC curve analysis of selected risk factors to predict in-hospital mortality after CABG.

RPR and RDW were independent risk factors selected by the multivariable logistic regression analysis. The red curve represents the prediction of RPR for in-hospital mortality, and the green curve illustrates the prediction of RDW for in-hospital mortality. Abbreviations: CABG, coronary artery bypass grafting; ROC, receiver operating characteristic; RPR, red blood cell distribution width-platelet ratio; RDW, red blood cell distribution width.

Table 3 Baseline characteristics between high and low RPR groups.

Variables	High RPR (n = 215)	Low RPR (n = 1043)	χ2/Z	p value	
Age (years)	63 (58, 66)	62 (57, 66)	−1.056	0.291	
Gender, male, n (%)	136 (63.26)	580 (55.61)	4.251	0.039	
BMI (kg/m2)	26.13 (21.91, 30.31)	25.84 (21.90, 29.58)	−0.719	0.472	
MAP (mm Hg)	98 (86.67, 108.33)	97.33 (86.67, 108)	−0.436	0.663	
NYHA, n (%)			3.462	0.326	
I	1 (0.47)	12 (1.15)			
II	146 (67.91)	759 (72.77)			
III	61 (28.37)	242 (23.20)			
IV	7 (3.26)	30 (2.88)			
Previous MI, n (%)	108 (50.23)	552 (52.92)	0.518	0.472	
Previous PCI, n (%)	50 (23.26)	278 (26.65)	1.068	0.301	
Previous stroke, n (%)	28 (13.02)	121 (11.60)	0.345	0.557	
Number of stenosed arteries	3 (3, 3)	3 (3, 3)	−0.125	0.901	
Number of grafts	4 (3, 4)	4 (3, 4)	−0.188	0.851	
SYNTAX score	31 (26, 35)	31 (25, 35)	−0.067	0.946	
EuroSCORE	4 (3, 6)	4 (3, 5)	−0.225	0.822	
Cardiopulmonary bypass, n (%)	87 (40.47)	449 (43.05)	0.487	0.485	
WBC (K/μl)	6.9 (5.8, 8.1)	7 (5.8, 8.1)	−0.372	0.710	
Platelet (K/μl)	112 (82, 126)	211 (179, 266)	−22.232	<0.001	
RDW (%)	15.2 (14.1, 17.1)	14.3 (13.5, 15.7)	−5.899	<0.001	
Hemoglobin (g/L)	138 (133, 141)	138 (134, 142)	−1.599	0.110	
RPR	0.14 (0.12, 0.18)	0.07 (0.05, 0.08)	−23.116	<0.001	
Creatinine (umol/L)	65 (59, 71)	63 (58, 69)	−2.131	0.033	
BUN (mmol/L)	7 (6.1, 7.8)	6.6 (5.8, 7.6)	−2.758	0.006	
Albumin (g/L)	43 (39, 46)	42 (38, 47)	−0.637	0.524	
CK-MB (U/L)	12 (10, 15)	13 (10, 15)	−0.781	0.435	
Troponin I (ng/ml)	0.37 (0.06, 1.98)	0.27 (0.06, 0.98)	−0.741	0.459	
BNP (pg/ml)	295 (112, 754)	262 (97, 706)	−1.094	0.274	
LVEF (%)	45 (39, 51)	46 (40, 51)	−0.825	0.409	
Mild to moderate MR, n (%)	39 (18.14)	157 (15.05)	1.291	0.256	
Mild to moderate AR, n (%)	14 (6.51)	83 (7.96)	0.524	0.469	
Mild to moderate TR, n (%)	9 (4.19)	40 (3.84)	0.059	0.809	
Mild to moderate PAH, n (%)	19 (8.84)	113 (10.83)	0.757	0.384	
Peripheral artery disease, n (%)	44 (20.47)	191 (18.31)	0.544	0.461	
Hyperlipemia, n (%)	20 (9.30)	108 (10.35)	0.216	0.642	
COPD, n (%)	29 (13.49)	142 (13.61)	0.002	0.961	
Diabetes, n (%)	100 (46.51)	498 (47.75)	0.109	0.741	
In-hospital mortality, n (%)	28 (13.02)	17 (1.63)	67.086	<0.001	
One-year mortality, n (%)	40 (18.60)	19 (1.82)	112.323	<0.001	
Notes.

Abbreviations AR aortic regurgitation

BMI body mass index

BNP B-type natriuretic peptide

BUN blood urea nitrogen

CK-MB creatine kinase-myocardial band

COPD chronic obstructive pulmonary disease

LVEF left ventricular ejection fraction

MAP mean arterial pressure

MI myocardial infarction

MR mitral regurgitation

NYHA New York Heart Association

PAH pulmonary arterial hypertension

PCI percutaneous coronary intervention

RDW red blood cell distribution width

RPR red blood cell distribution width-platelet ratio

TR tricuspid regurgitation

Survival and subgroup analysis

In the multivariable logistic regression model, high RPR correlated with elevated risks of in-hospital mortality (OR 6.097, 95% CI [2.308–16.104], p < 0.001) and one-year mortality (OR 6.395, 95% CI [2.610–15.666], p < 0.001) after adjusting for all included covariates. As depicted in the Kaplan–Meier survival curve, patients with low RPR show better one-year survival than those with high RPR (Fig. 3). As depicted in Figs. 4 and 5, subgroup analyses revealed that high RPR consistently had increased risks of in-hospital mortality and one-year mortality in each subgroup.

Figure 3 The one-year cumulative survival probability for high and low RPR groups.

The Kaplan–Meier curve depicts the cumulative probability of one-year mortality between high and low RPR, and the log-rank test (p value < 0.0001) is used to compare the survival difference between the two groups. The blue curve indicates the survival of patients in the low RPR group, and the red curve illustrates the survival of patients in the high RPR group. The shaded area represents the 95% confidence interval. The number of patients at risk is given for each subgroup every three months. Abbreviations: RPR, red blood cell distribution width-platelet ratio.

Figure 4 The forest plot for the association of high RPR and in-hospital mortality in subgroups.

The forest plot shows odds ratios and 95% confidence intervals obtained from binary logistic regression analyses, illustrating the relationship between high RPR and in-hospital mortality in each subgroup. The median of continuous variables was utilized for the division of each subgroup. Abbreviations: BNP, B-type natriuretic peptide; CI, confidence interval; LVEF, left ventricular ejection fraction; MAP, mean arterial pressure; OR, odds ratio; RDW, red blood cell distribution width; RPR, red blood cell distribution width-platelet ratio.

Figure 5 The forest plot for the association of high RPR and one-year mortality in subgroups.

The forest plot shows odds ratios and 95% confidence intervals obtained from binary logistic regression analyses, illustrating the relationship between high RPR and one-year mortality in each subgroup. The median of continuous variables was utilized for the division of each subgroup. Abbreviations: BNP, B-type natriuretic peptide; CI, confidence interval; LVEF, left ventricular ejection fraction; MAP, mean arterial pressure; OR, odds ratio; RDW, red blood cell distribution width; RPR, red blood cell distribution width-platelet ratio.

Discussion

Currently, few studies have discussed the relationship between RPR and in-hospital mortality in patients following CABG. ROC curves demonstrated better discrimination of RPR than RDW for in-hospital mortality when RPR was analyzed as a continuous variable. For patients undergoing CABG, we have calculated the cut-off value of RPR and revealed that high RPR was an independent risk factor for in-hospital mortality and one-year mortality. Moreover, Kaplan–Meier survival curves showed the better prognosis of the low RPR group, which verified that the cut-off value of RPR can be used to stratify patients’ short-term and mid-term clinical outcomes. Univariable and multivariable logistic regression analyses and subgroup analyses were performed in order to provide a reliable conclusion for the association of RPR and the prognosis of patients with CABG.

RDW has historically been used to diagnose anemia (Arkew et al., 2022). Recent evidence indicates that elevated RDW is significantly linked to poor prognosis in various diseases (Arkew et al., 2022). In our study, high RDW is also an independent risk factor for mortality, which is consistent with previous studies (Bozorgi et al., 2016a; Bozorgi et al., 2016b; Wasilewski et al., 2018). RDW has been confirmed to be significantly associated with a variety of inflammatory biomarkers and cytokines (Wang et al., 2021). The inflammatory response has negative impacts on bone marrow function, as well as disturbing erythropoietin generation and inhibiting iron metabolism (Wang et al., 2021; Wang & Hsu, 2021). Oxidative stress alters the half-life of red blood cells and releases more premature red blood cells into the peripheral circulation, subsequently leading to an increase in red blood cell heterogeneity (Bujak et al., 2015; Wang & Hsu, 2021). In addition, surgical revascularization can cause an inflammatory response and activation of the adrenergic system and thus affect the maturation of red blood cells, leading to increased RDW (Yao et al., 2023). A meta-analysis clarified that increased RDW could indicate an adverse prognosis in heart failure patients (Shao et al., 2015). Likewise, in an adult congenital heart disease cohort, RDW has been shown to be remarkably associated with cardiovascular events, independently of N-terminal pro-BNP and other prognostic biomarkers (Baggen et al., 2018). In addition, Hou et al. (2018) found that RDW has a positive correlation with frailty in elderly patients. Frailty also serves as a risk factor for mortality and hospital readmissions in heart failure (Zhang et al., 2018).

Platelet-dependent thrombosis is a key factor in the progression of coronary heart disease. Decreased platelet count is associated with increased risk of death, infarct size, and reinfarction rates (Dahlen et al., 2021; Ito et al., 2018; Roh et al., 2020). Low preoperative platelet levels may potentially increase the likelihood of postoperative bleeding (Gunertem et al., 2021). Consequently, this elevated risk of postoperative bleeding can lead to a higher mortality rate after surgery (Ezelsoy et al., 2020). In clinical practice, aggressive antiplatelet or anticoagulant treatment may increase the risk of hemorrhagic stroke (Xu & Peng, 2023). On the contrary, several studies have also shown that high platelet levels are also associated with adverse clinical outcomes (Malyszczak et al., 2020). Platelets mediate and initiate thrombotic occlusion of the entire coronary artery and aggregate in the circulation, resulting in impaired microcirculation and inducing myocardial ischemia during reperfusion (Roh et al., 2020). In this study, a univariate logistic regression analysis indicated that platelet was a risk factor. However, after using multivariate logistic regression analysis to correct other confounding factors, no significant association was found between platelet levels and in-hospital mortality for patients after CABG. We considered that RPR, a factor with more predictive value, was included in the multivariate logistic regression model, which was the main reason for the insignificant difference in the relationship between platelets and in-hospital mortality. Besides, there are still no unified and consistent criteria for the threshold of platelets in predicting clinical prognosis among published articles. Large cohort studies are required to further detect its cut-off value on prognosis.

Based on the above factors, this study mainly focuses on the ratio of RDW and platelet-RPR. RDW and platelets as readily gained indicators from routine laboratory examinations, are commonly used in clinical practice, which can be conveniently and dynamically monitored on the advantage of the rapidity and low cost. Besides, RDW is also an independent risk predictor of in-hospital mortality after CABG in the multivariate regression model. ROC analysis revealed the discrimination of RPR was significantly better than that of RDW, confirming that RPR is a better predictive indicator. On the basis of current research combined with our results, the cut-off value of RPR can clarify the risk stratification on the short-term and mid-term prognosis, and high RPR implies a poor prognosis for patients undergoing CABG. The cut-off value of preoperative RPR, instead of RDW or platelet, may provide better guidance for the assessment of patients’ states and clinical outcomes, as well as timely adjust the treatment strategy. The predicting value of RPR on early prognosis needs to be paid more attention to. The cut-off value of RPR needs external validation by using multicenter data to ensure its generalizability and clinical applicability. This study identified the prognostic value of preoperative RPR on mortality after CABG, and the underlying biological mechanism by which RPR affects patients’ prognosis remains to be elucidated in subsequent studies.

There exist several limitations in this study. This is a single-center retrospective study. Data from other medical centers are needed to further ensure the robustness and generalizability of our results. Potential bias and undetected factors might exist although univariable and multivariable logistic regression, and subgroup analyses were used to adjust the confounding factors. Only the relationship between RPR and mortality can be inferred on account of the nature of the retrospective study design. Randomized controlled trials or prospective cohort studies are needed to further explore the causal association between RPR and mortality for patients after CABG. Besides, the relationship between RPR levels measured at various time intervals and clinical outcomes also deserves additional investigation. Future studies should explore whether changes in RPR measurements over time are associated with disease progression or the effectiveness of treatment.

Conclusions

In conclusion, preoperative high RPR could serve as an independent risk predictor for in-hospital mortality and one-year mortality, which can be utilized to assess the prognosis of patients and can further provide guidance for the treatment in patients following CABG. Further studies are necessary to discover the mechanism of RPR on prognosis for patients undergoing CABG.

Supplemental Information

Supplemental Information 1 Raw data

Supplemental Information 2 Comparison of selected indicators between survivors and deaths in hospital

Abbreviations: BNP, B-type natriuretic peptide; LVEF, left ventricular ejection fraction; RDW, red blood cell distribution width; RPR, red blood cell distribution width-platelet ratio.

Abbreviations

AR aortic regurgitation

AUC The area under the receiver operating characteristic curve

BMI body mass index

BNP B-type natriuretic peptide

BUN blood urea nitrogen

CABG coronary artery bypass grafting

CK-MB creatine kinase-myocardial band

COPD chronic obstructive pulmonary disease

IQR interquartile range

LVEF left ventricular ejection fraction

Lowess locally weighted scatter plot smoothing

MAP mean arterial pressure

MI myocardial infarction

MR mitral regurgitation

NYHA New York Heart Association

OR odds ratio

PAH pulmonary arterial hypertension

PCI percutaneous coronary intervention

RDW red blood cell distribution width

ROC receiver operating characteristic

RPR red blood cell distribution width-platelet ratio

SD standard deviation

TR tricuspid regurgitation

Additional Information and Declarations

Competing Interests

Author Contributions

Human Ethics

Data Availability

The authors declare there are no competing interests.

Bufan Zhang conceived and designed the experiments, performed the experiments, analyzed the data, prepared figures and/or tables, authored or reviewed drafts of the article, and approved the final draft.

Yize Liu performed the experiments, analyzed the data, prepared figures and/or tables, authored or reviewed drafts of the article, and approved the final draft.

Jiyang Zuo performed the experiments, prepared figures and/or tables, authored or reviewed drafts of the article, and approved the final draft.

Tianxu Song analyzed the data, prepared figures and/or tables, authored or reviewed drafts of the article, and approved the final draft.

Naishi Wu conceived and designed the experiments, authored or reviewed drafts of the article, and approved the final draft.

The following information was supplied relating to ethical approvals (i.e., approving body and any reference numbers):

TEDA International Cardiovascular Hospital granted ethical approval to carry out the study within its facilities (No. 2022-0802-1).

The following information was supplied regarding data availability:

The raw measurements are available in the Supplementary File.

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
