# Peer review of "Association of red blood cell distribution width-platelet ratio with mortality after coronary artery bypass grafting"

_PeerJ, doi:10.7717/peerj.19472_

## Round 0.1 · original submission · Minor Revisions

Four reviewers have commented, please respond to all the comments

Reviewer 1 ·

Basic reporting

1-Clear and unambiguous, professional English used throughout.
2-The article includes adequate introduction and background. Relevant previous literature is appropriately referenced.
3-The structure of the article, figures and tables conform to an acceptable 'standard sections' format.
Figures are relevant to the content of the article, of sufficient resolution, and appropriately identified and labeled.
4-The submission is 'self-contained'. It contains all the results related to the hypothesis.

Experimental design

Although the relationship between hematological parameters and mortality in patients after CABG has been studied previously in the literature, it was supported by long follow-up periods and strong statistics in this study.
Methods described with sufficient detail & information to replicate.

Validity of the findings

no comment

Additional comments

This article is important for the scientific community because it is a study that contributes to the literature.
The title of the article is appropriate
The abstract section of the article is comprehensive and explanatory.
The subsections and structure of the article are appropriate because the abstract, introduction, conclusion and discussion sections are consecutive, sufficient, simple and understandable.
The article is appropriate for the article because the subject is written in understandable and simple language, the abstract is short and concise, the data is used adequately and the graphics are used.
The referances are sufficient

·

Basic reporting

Dear authors, Thank you for sending your original manuscript “Association of red blood cell distribution width-platelet ratio with mortality after coronary bypass grafting” to PeerJ Journal. I have learned and enjoyed reading it.
You are commended for your outstanding clinical results in coronary artery bypass grafting (CABG), which reported an in-hospital mortality rate of 3.58%. Your findings are noteworthy, as they identify a high baseline RPR ratio as an independent risk factor for both in-hospital mortality and mortality at the one-year follow-up after isolated CABG. The manuscript is well-written and easy to understand, featuring a current bibliography that provides further exploration of the topic. The high-quality figures and tables included are relevant and effectively complement the text.
The problem is clearly defined, with rigorous research conducted. The methods are thoroughly described, and the conclusions are concise, clear, and directly related to the original research question, supported by the results.
However, there are several doubts, questions, suggestions, and comments that I would like to share with you:
1- When were the samples taken for analysis of laboratory parameters?
2- Can you provide any preoperative risk scale for your population (STS-Score, EuroScore, etc.)?
3- Can you provide the mean number of grafts per patient operated on in your population?
4- How can ROC curve analysis be performed for in-hospital mortality with a variable such as previous stroke, which is a qualitative variable? I believe that ROC curve analyses can only be applied to quantitative variables, like the others in Figure 2.
5- I advise the authors to add one more table, in which they compare patients who die in the hospital (n= 45, 3.58%), versus patients who do not die in the hospital (N= 1213, 96%) with the same variables in Table 2.
6- The RPR ratio (RDW/platelets) can increase either due to a rise in the numerator (RDW) or a decrease in the denominator (baseline platelets).
An elevation in the RDW is often associated with anemia. Have you investigated whether patients who died in the hospital had lower baseline hemoglobin levels compared to those who survived?
From your findings, it can be inferred that lower baseline platelet levels may have worse outcomes. The authors should address this observation in their discussion.
7- Have you studied the RPR relationship in the postoperative period of your patients?
8- Figure 5, which shows the subgroup analysis of one-year mortality in the study population, includes all patients in the study (1258) when only 1215 patients should be included in this analysis, once patients who died in the hospital (45) have been excluded. The authors should review this analysis.
9- What was the percentage of patients lost to follow-up?

Experimental design

The manuscript is well-written and easy to understand, featuring a current bibliography that provides further exploration of the topic. The high-quality figures and tables included are relevant and effectively complement the text.
The problem is clearly defined, with rigorous research conducted. The methods are thoroughly described, and the conclusions are concise, clear, and directly related to the original research question, supported by the results.

Validity of the findings

The conclusions are concise, clear, and directly related to the original research question, supported by the results.

Additional comments

None

Reviewer 3 ·

Basic reporting

The article is well written and is relevant to the field.
A suggestion is to add more flow to the DISCUSSION part where a lot of references , although well cited may make it difficult for the international audience to keep up with the information being presented.

For example in line 180 "I n our study, high RDW is also an independent risk factor for mortality, which is consistent with.." . This line could have started by first describing the context of the studies cited then adding the authors' findings.

Experimental design

No comment

Validity of the findings

No comment

Additional comments

I congratulate the authors . The work shows great promise ,likely, in identifying predictors of morbidity/mortality after CABG. A modest touch up of the discussion part will positively add to the impact of this study.

·

Basic reporting

Reviewer Comment For Author: Association of red blood cell distribution width- platelet ratio with mortality after coronary artery bypass grafting


Support criticisms with evidence from the text or from other sources. With specific suggestions on how to improve the manuscript

- line 31: The cut-off value of RPR 0.10679 has been exaggerated to be of five digits, I think this is a mathematical manipulation. I suggest to be 2or 3 digits’ maximum
- line 126-127: 3.58%. of 1258 patients is the incidence of in-hospital. mortality is very low in my Opinion. It needs more Comments
- line 96: RDW and platelet count, were they measured by the same call counter? Machines vary in results and they need Strict control
Line 77 and 78: The study was done in one medical center. I suggest to do it on patients from different medical centers
line 227-229: I Stress that RPR needs further investigation
line 243: The RPR mechanism on prognosis for patients undergoing CABG has not been discovered clearly. Thus , I consider this study is just theoretical and mathematical
The abstract without keywords

Experimental design

not found, this study retrospectively

Validity of the findings

The findings of the study appear valid based on the presented methodology and statistical analysis.

Additional comments

The manuscript titled "Association of Red Blood Cell Distribution Width-Platelet Ratio with Mortality After Coronary Artery Bypass Grafting" addresses an important and clinically relevant topic. The study is well-conducted, and the findings are valuable for clinicians managing patients undergoing coronary artery bypass grafting (CABG). The focus on red blood cell distribution width-platelet ratio (RPR) as a potential biomarker for mortality adds to the growing body of evidence regarding the prognostic value of hematological indices in cardiac surgery.

The manuscript is generally well-structured, and the methodology is clearly explained. The data analysis is appropriate, and the conclusions align with the presented results. However, there are a few areas that require clarification or improvement to enhance the clarity and impact of the paper.

Minor Revisions

---

## Round 0.2 · Major Revisions

- Abbreviations should be explained at first appearance
- Methods are very poor. In particular, mention laboratory equipment, manufacturers etc.; New York Heart Association (NYHA) is no data (!) do you mean classification? ; How were all these data assessed?
- Figure qualities are poor. Fig 1 CI? Fig 3 layout; Fig4&5 x.axes surplus points (?)
- Figure legends are poor as well
- Red blood cell distribution width-platelet ratio is in the title. It is not explained.

·

Basic reporting

unambiguous professional English is used throughout.
Literature references, and sufficient field background/context are provided.
Professional article structure, figures, tables. Raw data shared.
The manuscrit is Self-contained with relevant results to hypotheses.

Experimental design

The researchh question i well-definedd, relevant & meaningful. It is stated how research fills an identified knowledge gap
Original primary research within the Aims and Scope of the journal.
Rigorous investigation performed to a high technical & ethical standard.
Methods described with sufficient detail & information to replicate.

Validity of the findings

Impact and novelty not assessed. Meaningful replication encouraged where rationale & benefit to literature is clearly stated
All underlying data have been provided; they are robust, statistically sound, & controlled
Conclusions are well stated, linked to original research question & limited to supporting results

Additional comments

I want to thank the authors for agreeing with the suggestions made by the reviewers and for including them in the manuscript, which I believe will greatly interest the entire scientific community concerned with this topic.

Reviewer 3 ·

Basic reporting

Meets the required criteria

Experimental design

Meets the required criteria

Validity of the findings

Meets the required criteria

Additional comments

The authors have done a commendable job in revising the manuscript as per the suggestions of reviewers. Paper can be published .

·

Basic reporting

clear

Experimental design

accepted

Validity of the findings

accepted

Additional comments

no

---

## Round 0.3 · accepted · Accept

Authors have addressed all my comments.